# Breast Cancer Metastasis: Mechanisms and Therapeutic Implications

**DOI:** 10.3390/ijms23126806

**Published:** 2022-06-18

**Authors:** Misung Park, Dohee Kim, Sunghyub Ko, Ayoung Kim, Kyumin Mo, Hyunho Yoon

**Affiliations:** 1Department of Medical and Biological Sciences, The Catholic University of Korea, Bucheon 14662, Korea; ms8006@catholic.ac.kr (M.P.); dh0864@naver.com (D.K.); koshmbs0515@gmail.com (S.K.); ayoung_0615@naver.com (A.K.); reotung12@naver.com (K.M.); 2Department of Biotechnology, The Catholic University of Korea, Bucheon 14662, Korea

**Keywords:** breast cancer, metastasis, tumor microenvironment, EMT

## Abstract

Breast cancer is the most common malignancy in women worldwide. Metastasis is the leading cause of high mortality in most cancers. Although predicting the early stage of breast cancer before metastasis can increase the survival rate, breast cancer is often discovered or diagnosed after metastasis has occurred. In general, breast cancer has a poor prognosis because it starts as a local disease and can spread to lymph nodes or distant organs, contributing to a significant impediment in breast cancer treatment. Metastatic breast cancer cells acquire aggressive characteristics from the tumor microenvironment (TME) through several mechanisms including epithelial–mesenchymal transition (EMT) and epigenetic regulation. Therefore, understanding the nature and mechanism of breast cancer metastasis can facilitate the development of targeted therapeutics focused on metastasis. This review discusses the mechanisms leading to metastasis and the current therapies to improve the early diagnosis and prognosis in patients with metastatic breast cancer.

## 1. Introduction

Breast cancer, the most common cancer in women, develops in the breast lobules, tubes, or connective tissue. Although various criteria and classifications are possible, the most representative types of breast cancer can be divided into four types based on molecular subtypes: luminal A, luminal B, human epidermal growth factor receptor 2 (Her2) positive, and triple negative breast cancer (TNBC). These subtypes have very different metastases, prognosis, and treatment methods [1]. In terms of molecular targeting approaches to breast cancer, targeting Her2, a tyrosine receptor kinase, has been widely used in a relatively successful manner for Her2-positive breast cancer [2,3,4]. Her2 is reported to be overexpressed in 20–25% of breast cancers, which is associated with cancer cell aggressiveness [5,6,7]. TNBC, the most aggressive breast cancer, does not express estrogen receptor (ER), progesterone receptor (PR), and Her2. TNBC accounts for only 15–20% of breast cancers. However, it is usually identified at an advanced stage at the time of diagnosis, resulting in a high recurrence rate and low survival rate [8,9]. Additionally, TNBC has no effective targeted therapy and most commonly metastasizes to the brain, bones, lungs, and liver [10]. Recently, various FDA-approved drugs targeting primary/metastatic breast cancer have been developed and applied to patients, but they still have many limitations (Table 1).

Metastasis is the multiple process by which an original primary tumor develops into a distal secondary tumor. It is a representative hallmark of cancer and leads to treatment failure, leading to the death of many patients [25]. Therefore, the patient’s prognosis is closely related to metastasis. The diagnosis of metastatic cancer is considered the final stage in most cancer types. Metastasis is highly complex and involves multiple cellular mechanisms including division from the primary tumor, invasion, evasion of immune surveillance, and regulation of the tissue microenvironment. In particular, epithelial–mesenchymal transition (EMT) is required for metastasis in most cancers [26].

Ovarian cancer metastasis proceeds through a mechanism in which cancer cells are transported away from the primary tumor by the physiological movement of peritoneal fluid into the peritoneum and reticulum. As a result, ovarian cancer metastasizes to the abdominal cavity, omentum, and parenchyma of the liver or lungs [27]. However, in breast cancer, metastasis to the central nervous system (CNS), either brain parenchyma or leptomeninges, corresponds to a late feature of progressive metastatic disease. CNS metastases occur in 10–15% of patients with metastatic breast cancer [28], associated with poor clinical outcomes.

EMT, a phenomenon in which epithelial cell polarity and intercellular cohesion are lost, is required for the initiation of metastasis in breast cancer [29]. The pleiotropic transcription factors including SNAIL, TWIST, and zinc finger E-box-binding (ZEB) are important molecules in the regulation of EMT. Recent studies have demonstrated that the loss of expression or dysfunction of GATA3 promotes breast cancer metastasis by regulating G9A and metastatic tumor antigen (MTA) family proteins recruited by ZEB2 [30]. Inflammatory cytokines including interleukin-6 (IL-6) play essential roles in breast cancer metastasis. IL-6 binds to the IL6R/gp130 complex and activates downstream of the Janus kinase (JAK) signals to activate signal transducers and activators of transcription 3 (STAT3) [31]. In turn, IL-6/STAT3 signaling promotes EMT by regulating estrogen receptor α (ERα), which is essential for breast cancer metastasis [31]. In addition, cell surface proteins play an important role in breast cancer metastasis. Interactions between cancer cells and the cell surface mediate cell adhesion and invasion to initiate metastasis [32]. For example, metatherin, overexpressed in more than 40% of breast cancers, binds to the pulmonary vasculature by the C-terminal segment of the extracellular domain [32], leading to poor clinical outcomes in breast cancer through metastasis.

Immune cells and the tumor microenvironment (TME) may also contribute to breast cancer metastasis. Tumor-associated macrophages (TAMs) play different roles in various microenvironmental signaling. In breast cancer, TAM, a macrophage subtype M2, is often activated by IL-4 secreted by CD4+ T cells [33]. TAM-secreting cytokines including chemokines, inflammatory factors, and growth factors are strongly associated with metastasis by increasing the adhesion to the extracellular matrix (ECM) [33]. Therefore, these immune cells and secreted factors could be targets for the treatment of metastatic breast cancer.

Metastasis is important in breast cancer treatment because it can be both a therapeutic target and a detectable marker. Targeted therapeutics for breast cancer include substances or drugs that block cancer growth by interfering with the function of certain molecules responsible for tumor cell proliferation and survival. Targeted treatment for metastatic breast cancer is determined by the presence or absence of hormone receptors, Her2, cancer recurrence, metastasis rate, and metastasis site [34]. In particular, personalized treatment for individual patients can enhance the therapeutic efficacy and minimize chemotherapy-induced toxicity in breast cancer [35]. However, the possibility of recurrence and metastasis of all types of breast cancer still exists. Additionally, there is still no standard treatment for patients with TNBC, the most aggressive and recurrent type, and results in intensive metastasis. Therefore, this review highlights recent advances in breast cancer treatment, clinical potential and limitations, and presents a more in-depth knowledge of metastatic breast cancer.

## 2. EMT in Metastatic Breast Cancer

### 2.1. Relationship between EMT and Metastasis

EMT is the process by which cells with epithelial characteristics are transformed into mesenchymal cells, one of the main mechanisms involved in cancer metastasis. It is reversible because mesenchymal cells can redifferentiate into epithelial cells or other cells [36]. EMT and its reverse process, mesenchymal-to-epithelial transition (MET), occur throughout wound healing, fibrosis, and tumor progression. The specific set of regulatory changes that enable EMT induces the normal process of increased differentiation in developing cell populations within an organism [37]. However, during EMT, cancer cells acquire the ability to migrate and become active as they progress to malignancy [38]. The EMT program converts epidermal cells with strong intercellular bonds into migrating mesenchymal cells so that cancer cells can invade other areas. Cancer cells migrating through blood vessels and lymphatic vessels reach other tissues and undergo MET, the reverse process of EMT, in which they are converted back into epithelial cells with tight bonds in the distant sites. Additionally, EMT-derived tumor cells are resistant to treatment due to their stem cell properties. Therefore, several therapies targeting EMT for patients with metastases are being developed [39].

### 2.2. EMT Transcription Factors (EMT-TFs) in Metastatic Breast Cancer

Transcription factors (TFs) control gene expression processes by binding to chromatin, which plays an essential role in cancer progression and metastasis [29]. EMT-TFs can be classified according to their ability to directly or indirectly inhibit E-cadherin. E-cadherin is an epithelial cell adhesion protein involved in tumor metastasis by allowing the invasion of cancer cells. Specifically, the loss of E-cadherin is associated with the upregulation of growth factor-β (TGFβ), reactive oxygen species, and genes involved in altering apoptosis signaling pathways [40]. The inhibition of E-cadherin led to increased mesenchymal cell markers including N-cadherin and vimentin [41]. The TWIST1, SLUG, SNAIL, ZEB1 (TCF8/dEF1), ZEB2 (SIP1), and FOX families have been found to be transcriptional inhibitors of E-cadherin. These EMT-TFs are strongly associated with cancer initiation, progression, invasion and metastasis as well as resistance to treatment by regulating various gene expression through various combinations [42].

Overexpression of EMT-TFs in breast cancer is a prognostic factor in metastatic breast cancer and contributes to cancer development and progression. For example, TWIST1 expression was highly associated with hypermethylation and hypoacetylation of the E-cadherin promoter by recruiting TWIST1 and Mi2/nucleosome remodeling and deacetylase (Mi2/NuRD) protein complexes in metastatic breast cancer [41]. The TWIST1/Mi2/NuRD protein complex is inhibited by E-cadherin expression to promote EMT to induce breast cancer metastasis, demonstrating that knockdown of TWIST1 plays an essential role in the inhibition of metastatic breast cancer [41]. SNAIL also inhibits the expression of E-cadherin by binding to the promoter of E-cadherin, which increases the expression of vimentin, leading to the EMT process [43]. Dong et al. found that SNAIL interacts with Suv39H1 (repressor of variant 3–9 homologue 1), a major methyl-transferase responsible for H3K9me3, contributing to breast cancer metastasis through epigenetic regulation [44].

Similarly, the ZEB family represses the expression of E-cadherin and recruits other chromatin modulators to enhance the expression of vimentin and N-cadherin. Whole genome analysis demonstrated that ZEB1-mediated activating protein-1 (AP-1) and Yes1 Associated Transcription Regulator (YAP1) activate tumor-promoting genes to induce aggressive breast cancer types [45]. These studies provide a promising therapeutic option to target these EMT-TFs or mediators in patients with metastatic breast cancer.

### 2.3. EMT-Mediated Cancer Stem Cells in Metastatic Breast Cancer

Current studies have shown that cells with stem cell characteristics, capable of self-renewal and differentiation into other cell types, make significant contributions to tumor initiation, progression, and metastasis [46,47]. Although the origin of cancer stem cells is largely unclear, biomarkers of breast cancer stem cells (BCSCs) have been relatively well validated. For instance, a sub-set of breast cancer cells with CD44^+^/CD24^−/low^ have a more aggressive phenotype, resulting in drug resistance and breast cancer metastasis [48,49]. In addition, increased aldehyde dehydrogenase 1 (ALDH1) activity promotes self-renewal and tumor initiation ability, which may be a predictive marker for poor clinical outcome in breast cancer [50]. These studies suggest that cells with CD44^+^/CD24^−/low^ and ALDH1 are widely considered as markers of BCSCs.

BCSCs significantly exhibited improved the self-renewal and tumor initiation capacity as well as the high expression of EMT markers such as SLUG, TWIST1, and FOXF2 [51]. Storci et al. found that SLUG improved the stability of cytoplasmic β-catenin by inducing the expression of tumor necrosis factor-α (TNF-α) and IL-8. In turn, miR-221 bound to β-catenin reduced Rad51 and ERα targeting, maintaining a pro-inflammatory phenotype in BCSCs [52]. TWIST1 expression in breast cancer has been shown to induce cancer stem cell characteristics. One study showed that TWIST1 binds to the promoter of CD24 and transcriptionally regulates CD24 expression, resulting in stem cell traits such as self-renewal and the ability to form mammospheres in breast cancer [53], suggesting that EMT-TFs are mainly involved in the BCSC phenotype.

CSC transcription factors (CSC-TFs) are considered as important regulators in the initiation of a metastatic niche that includes crosstalk between tumor-derived factors and terminal stromal components. Lin28, an RNA-binding protein, is one of the master regulators of embryonic stem cell self-renewal that correlates with clinical tumor grade and the terminal metastasis of breast cancer [50]. In the MMTV-PyMT mouse model, Lin28B did not affect the primary tumor growth but promoted the development of lung metastases [54], suggesting that the ability of cancer cells to self-renew can promote distant metastasis in breast cancer. IL-6 secreted from differentiated cancer cells supports CSC survival, metastasis, and treatment resistance in breast cancer [55,56]. The transcription factor CCAAT/enhancer binding protein δ (C/EBPδ) was highly overexpressed in normal breast epithelium and in low-grade, hormone receptor-positive breast cancers [57]. Balamurugan et al. found that C/EBPδ recruited by hypoxia and IL-6 directly targeted the promoter of CSC-TFs, contributing to CSC maintenance and breast cancer metastasis [56]. Thus, the study showed that the combination of metformin and conventional chemotherapy, which selectively inhibits CSCs, may be a better strategy to treat metastatic breast cancer.

### 2.4. EMT-Targeted Therapy in Metastatic Breast Cancer

EMT determines the most lethal properties and chemical resistance of metastasis formation, making it an attractive target for cancer therapeutics. Pirfenidone, an anti-fibrotic and anti-inflammatory agent, reduces fibroblast proliferation by inhibiting the transforming growth factor-β (TGF-β) signaling pathway, a key EMT regulatory signal [58,59]. In a murine breast cancer model, pirfenidone suppressed tumor burden by targeting downstream of TGF-β such as SMAD3, p38, and AKT [60]. In addition, pirfenidone interfered with Hedgehog pathway signaling, a valuable additional benefit during breast cancer treatment [61]. Galunisertib, a small molecular inhibitor of the kinase domain of the TGF-β receptor I, explicitly decreases the phosphorylation of SMAD2 (p-SMAD) [62]. Holmgaard et al. showed that galunisertib treatment inhibited TGF-β-induced p-SMAD and reversed CD8+ T cell-mediated immune suppression [63]. This study provided a better strategy for treating breast cancer that could enhance the anti-tumor effects in combination with programmed cell death 1 receptor ligand (PD-L1)/PD-1 checkpoint inhibitors and galunisertib [63]. Metformin is known to increase 5′ adenosine monophosphate-activated protein kinase (AMPK) levels and activity, a key regulator of metabolism and protein synthesis [64]. Activated AMPK inhibited mTOR and the expression of β-catenin in breast cancer cells [65], which blocks the translocation of β-catenin to the nucleus to promote the expression of EMT-related genes [66]. Taken together, metformin therapy as an EMT blocker may be used in patients with metastatic breast cancer. The Wnt/β-catenin signaling pathway is implicated in various kinds of biological processes in cancer including EMT [67,68]. Baicalein has been found to inhibit the expression of Wnt1 and β-catenin, leading to suppressed Cyclin D1 and Axin2 expression [69]. As a result, baicalein markedly reduced breast cancer metastasis by downregulating Wnt/β-catenin-mediated EMT [69]. These studies suggest that targeting EMT pathway signaling or EMT-linked genes can yield effective results in the treatment of breast cancer metastasis.

## 3. Tumor Microenvironment (TME) in Metastatic Breast Cancer

### 3.1. TME-Associated Cells in Metastatic Breast Cancer

Tumors are heterogeneous diseases and consist of many other cells in addition to the surrounding tumor cells. Interactions with these cells that make up surrounding tumors result in cancer proliferation and metastasis [70]. These cellular and peripheral signals are collectively referred to as the TME, consisting of non-cellular substances such as ECM and cytokines and cellular components such as cancer-associated fibroblasts (CAFs), immune cells, endothelial cells, and adipocytes. These diverse components of TME function in cancer initiation, progression, metastases, and treatment resistance [71].

TME is composed of various types of immune cells. Anti-tumor immune cells such as cytotoxic CD8+ T lymphocytes and natural killer (NK) cells interact with antigen-presenting cells (APCs) that display antigens bound by major histocompatibility complex (MHC) proteins on their surface for cytotoxic responses in the TME [72,73]. Some tumor-promoting immune cells such as M2 macrophage and myeloid-derived suppressor cells (MDSCs) were recruited by inflammatory cytokines including CCL2 in the TME, which enhances cell proliferation, vascularization, migration, and migration metastasis [74,75].

TME is composed of not only immune cells but also non-immune cells such as CAFs, adipocytes, and endothelial cells, which leads to promoting the progression and invasion of the tumor by secreting growth factors and pro-inflammatory mediators, cell–cell interactions, and metabolic crosstalk with tumor cells [76]. In breast cancer, tumor cells co-cultured with CAF showed increased expression of IL-6 and IL-8, leading to more invasive and angiogenic capacity [77]. Adipocytes have been reported to be associated with an epidemiologic association between obesity and postmenopausal breast cancer risk in breast cancer development and progression [76,78]. In addition, IL-6 derived from cancer-associated adipocytes induced breast cancer invasion and metastasis by propagating cancer stem cell expansion [79,80,81]. These studies have demonstrated that TME-associated cells are essential for breast cancer metastasis.

### 3.2. Notch Signaling in the TME of Metastatic Breast Cancer

The Notch pathway not only regulates the self-renewal of normal mammary stem cells, but also plays an essential role in promoting breast cancer initiation, tumor progression, cell maintenance, tumor cell fate, proliferation, survival, and motility [82]. In breast cancer, Notch has the ability to self-renew and may act as a regulator of tumor-initiating cells (TICs) that repopulate tumor cells [83,84,85]. Non-malignant cells in the TME of breast cancer such as vascular endothelial cells, immune cells, and CAFs support the activation of the Notch pathway by interactions between non-malignant and malignant components [86]. For example, ADAM10 secreted by CAF increased the expression of ALDH, a biomarker of breast cancer stem cells, which promotes cancer cell proliferation, motility, and survival in breast cancer by regulating the downstream of Notch signaling [87]. In addition, Notch directly regulated *CCND1* genes that play essential roles in the G1 phase of the cell cycle, which has an important function as a pathogenic effector in breast cancer [88,89]. During angiogenesis, the interaction between Notch and vascular endothelial growth factor (VEGF) in the TME of breast cancer induced breast cancer metastasis by activating VEGF receptors [90]. Notch signaling is also associated with ECM remodeling, which is normally remodeled by matrix metalloproteinases (MMPs), an essential step in metastatic breast cancer. In detail, the NF-kB pathway activated by Notch signaling increased MMP-2 and MMP-9 genes, leading to the activation of urokinase PA (uPA), a plasminogen activator, which causes severe symptoms or death in breast cancer [89]. These studies suggest that the development of Notch-targeted therapeutics may be a better strategy for the treatment of metastatic breast cancer.

### 3.3. TME-Targeted Therapy in Metastatic Breast Cancer

Since stromal cells in the TME of breast cancer have contributed to breast cancer stemness, therapeutic resistance, and metastasis, TME is importantly considered a druggable target. TAM, a type of immune cell, play essential roles in the tumor progression and metastasis of breast cancer. The infiltration and differentiation of TAMs regulate the anti-tumor immune responses by the secretion of various cytokines and chemokines, leading to poor prognosis [91,92,93]. Both the selective monocyte targeting chemotherapeutic agent, trabectedin, and colony-stimulating factor 1 (CSF1) inhibitor that targets TAM can inhibit the recruitment of macrophages to TME, resulting in reduced tumor growth and metastasis formation [94,95]. Adipose mesenchymal stem cells can differentiate into mature adipocytes, which promote the migration of breast cancer cells through cytokine signaling. Given the observations, sulforaphane (SFN, a dietary chemoprevention agent) was treated in the co-culture adipose mesenchymal stem cells with breast cancer cells, resulting in decreased adipogenic differentiation through the maintenance of MSC self-renewal, thereby reducing breast cancer cell migration and tumor formation [96].

As above-mentioned, Notch signaling is a promising therapeutic target in patients with metastatic breast cancer. Treatment with a γ-secretase inhibitor (PF-03084014, Pfizer, New York, NY, USA; MK-0752, MERK, Rahway, NJ, USA), a potent Notch inhibitor, significantly reduces tumor burden in metastatic TNBC through G0/G1 arrest [97,98]. In addition, BXL0124, a Gemini vitamin D analog, was evaluated in basal-like breast cancer cells, resulting in decreased CD44^+^/CD24^−/low^ population by inhibiting Notch1 signaling [99]. Notch depletion may improve clinical prognosis by targeting tumor-initiating cells in metastatic breast cancer.

## 4. Epigenetic Role in Metastatic Breast Cancer

### 4.1. Regulation of DNA Methylation in Metastatic Breast Cancer

Epigenetic changes are closely related to breast cancer metastasis. Epigenetics is formally defined as heritable changes in gene expression or chromosomal stability. Epigenetic mechanisms are required for the normal development and maintenance of tissue-specific gene expression in humans [100]. However, this process is deregulated in cancer. Although epigenetic states are ‘well-balanced’ in normal cells, they can be altered in a variety of ways in cancer cells. Epigenetic abnormalities in breast cancer can shift the cellular state from normal to malignant, leading to tumor initiation, progression, and metastasis [101]. Important epigenetic changes include DNA methylation, acetylation, and histone modifications of target genes.

DNA methylation is the addition of a methyl group to the cytosine of a CpG dinucleotide by DNA methyltransferase (DNMT). CpG islands are referred to as CpG clusters, which are clear targets for DNA methylation located at the 5′ end of human genes. In most cases, all CpG islands remain unmethylated. It has been reported that DNMT1 and DNMT2B were highly expressed in breast cancer, contributing to cancer progression and metastasis through DNA methylation [102,103]. Methylated DNA attracts a methyl-CpG binding protein, namely histone deacetylase (HDAC), causing histone modifications and gene expression [104]. Methylation of tumors suppresses genes that contribute to cancer metastasis. In general, methylation causes the repression of gene expression, whereas demethylation makes gene expression reactive [105]. In breast cancer, the *APC*, *CDH1*, and *CTNNB1* genes are highly methylated [106]. High levels of methylation serve as operable detection markers in breast cancer patients. This suggests that the methylation of tumor suppressor genes is involved in the metastasis and progression of breast cancer.

The *LHX2* gene exhibited significant methylation changes at two different CpG sites in breast cancer [107]. Some breast cancer risk-associated genes such as *LHX2*, *TFAP2B*, *JAKMIP1*, and *SEPT9* were differentially methylated in site-specific and differentially methylated region (DMR) analysis. This gene can be expected to increase risk stratification to develop new strategies for the prevention and treatment of breast cancer [107]. Li et al. found that PRMT1-mediated EZH2 (a potentiator of essential histone methyltransferase and EMT inducer, zeste homolog 2)-R342 methylation enhanced EZH2 stability, promoting breast cancer cell metastasis. In the study, EZH2 can undergo symmetric dimethylation whereas the EZH1 protein does not undergo asymmetric dimethylarginine or symmetric dimethylation modifications [108]. The importance of R342-EZH2 methylation in increasing breast cancer metastasis is emerging and has clinical value as a biomarker for breast cancer diagnosis and treatment.

### 4.2. Regulation of Histone Modification in Metastatic Breast Cancer

Histone modifications regulate the structure of chromatin and alter the accessibility of DNA. Histone modifications such as acetylation, phosphorylation, and methylation are vital factors. It is involved in significant histone modifications by enzymes including histone acetyltransferase (HAT), HDAC, and histone methyltransferase (HMT). HAT is described as a “writer” and HDAC is described as an “eraser” [105]. At different locations and types, histone modifications can affect gene expression including activation or repression. For example, reduced levels of lysine acetylation (H3K9ac, H3K18ac, H4K12ac) and methylation (H3K4me2, H4K20me3) and arginine methylation (H4R3me2) correlated with poor prognosis in breast cancer [109]. Mainly, histone acetylation induces gene activation, and histone methylation induces gene activation or repression depending on the modification site.

Histone methylation refers to the transfer of a methyl group from S-adenosylmethionine to a lysine or arginine residue by HMT, and histone demethylase (HDM) removes the methyl group. Most HATs are controlled by domain binding, post-translational modifications, or acetylation. Recent studies have shown that HDAC2 was essential for increasing breast cancer cell motility through the induction of the breast cancer metastasis markers MMP2 and N-cadherin [110]. Based on these results, four HDAC inhibitors have been approved by the FDA for the clinical treatment of peripheral T-cell lymphoma, cutaneous T-cell lymphoma, and multiple myeloma [110]. In terms of histone methylation, PRMT5 is a representative target for breast cancer therapy. Several studies have reported that the inhibition of PRMT5 in breast cancer can promote a decrease in metastatic capacity and proliferation. Therefore, HMT inhibitors are being evaluated preclinically by targeting PRMT3, PRMT4, PRMT5, and PRMT6, which exhibit tumor suppression [111].

Histone phosphorylation of H2B and H3 was reported to contribute to DNA repair, mitosis, and gene regulation [112]. H3 phosphorylation promoted chromosome condensation during mitosis, leading to cell proliferation. H3 phosphorylation also has part of a signaling cascade that promotes breast cancer metastasis [113]. In terms of the relationship between histone phosphorylation and subtypes of metastatic breast cancer, histone acetylation H3K9ac was found to correlate with HER2-positive and TNBC subtypes [113]. Thus, breast cancer cells with different phenotypes display different patterns of histone variations, which have different effects on gene expression.

### 4.3. Epigenetic Change-Targeted Therapeutic Strategies for Metastatic Breast Cancer

Epigenetic alterations could be therapeutic targets for breast cancer that small molecules could target. Some therapies targeting epigenetic changes such as EZH2, IDH, HDAC, and DNMT inhibitors are already FDA-approved and are undergoing clinical trials to treat solid (NCT01928576 and NCT03179943) and hematological tumors (NCT03164057 and NCT022717884) [101].

DNMT1 is an epigenetic target for breast cancer cell damage and alters metastatic and aggressive phenotypes. DNMT1 performs oncogenic functions in breast cancer by inhibiting ER expression, inducing EMT, promoting cell autophagy, and increasing CSCs [114]. This suggests the need for DNMT1 to promote metastasis and invasion in breast cancer. Therefore, novel DNMT1 inhibitors such as azacitidine, decitabine, and guadecitabine, which can make cancer patients sensitive to immune checkpoint blockade therapy, have shown potential antitumor effects in breast cancer cells [114].

A member of the Spalt-like transcription factor family, SALL2 plays an essential role as an upstream regulator of ER expression. It also directly regulates the expression of related genes, resulting in apoptosis, growth arrest, and the maintenance of quiescence by the regulation of gene expression [115]. Interestingly, *SALL2* hypermethylation was associated with decreased disease-free survival in tamoxifen-resistant breast cancer, suggesting that SALL2 may have a dual function in breast cancer cells. This study provides evidence that the combination of a DNMT inhibitor and tamoxifen may be an effective treatment for some breast cancer patients [116]. Taken together, epidrugs may support the opportunity to replace therapeutically inactive cells with sensitive ones.

## 5. Conclusions and Perspective

Breast cancer is a heterogeneous disease where the treatments are not limited to one (Figure 1). Targeted therapies such as trastuzumab and pertuzumab are generally used in combination with other anticancer drugs. In the HER2-positive subtype, various therapeutics targeting HER2 have been developed. In fact, trastuzumab or docetaxel plus pertuzumab was effective in metastatic breast cancer [117]. In addition, significant results were obtained when tucatinib was used in combination with trastuzumab and capecitabine for patients with HER2-positive metastatic breast cancer [118]. In particular, TNBC with BRCA1 mutation responds to PARP inhibitors [119]. However, the possibility of the recurrence and metastasis of all types of breast cancer still exists.

Despite recent medical advances, metastasis remains the leading cause of death in breast cancer patients. Through intensive research, various mechanisms leading to breast cancer metastasis have been elucidated (Figure 2), and drugs have been developed to inhibit these mechanisms. However, it still cannot prevent the death of patients from metastasis. This is because breast cancer metastasis is not induced by a single mechanism, but is comprehensively caused by various mechanisms. Recent studies have shown that anti-PD-L1 therapeutic antibodies such as pembrolizumab or atezolizumab have significant antitumor activity in metastatic breast cancer [120,121,122], suggesting that immune checkpoints significantly alter the metastatic cascade of breast cancer. Many studies have shown that no single drug treatment can permanently remove the tumor. Therefore, along with existing treatments for metastatic breast cancer, immune checkpoint blockade can be a promising treatment strategy, and clinical treatment trials are urgently needed.

## Figures and Tables

**Figure 1 ijms-23-06806-f001:**
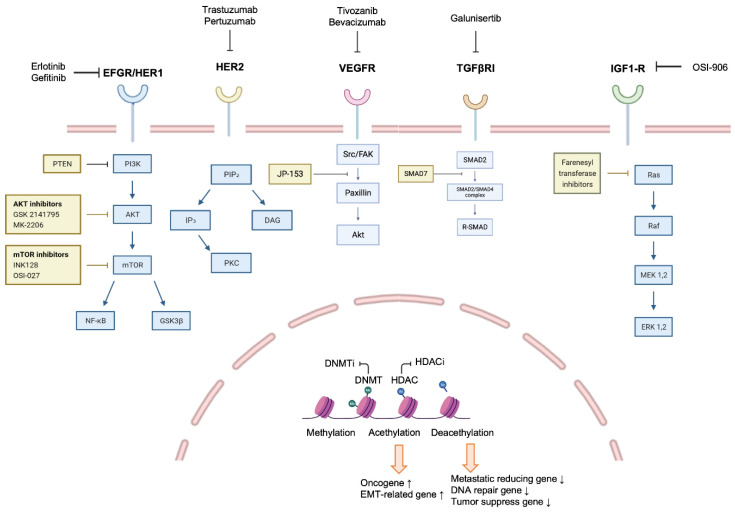
The general regulatory signaling and related inhibitors associated with breast cancer. The mechanisms through various receptors such as EGFR, HER2, VEGFR, TGFβR, and IGF1-R are closely associated with breast cancer initiation, growth, and metastasis. In addition, epigenetic changes also induce aggressive properties of breast cancer to induce metastasis. Therefore, these complex mechanisms can be a target for metastatic breast cancer treatment, and small molecular inhibitors and monoclonal antibodies that inhibit these mechanisms have been developed.

**Figure 2 ijms-23-06806-f002:**
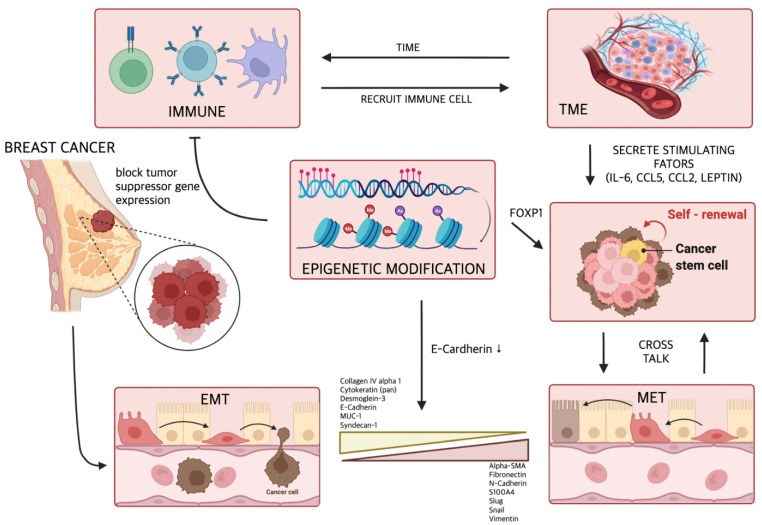
The various mechanisms and factors inducing the metastasis of breast cancer. Induction of EMT is directly related to the metastasis of breast cancer. In addition, multiple cytokines secreted by TME and epigenetic modification promote breast cancer metastasis by maintaining and expanding cancer stem cells with drug resistance and metastatic potential.

**Table 1 ijms-23-06806-t001:** The FDA-approved drugs in primary and metastatic breast cancer.

Primary/Metastasis	Drug Name	Subtype	Effect	Ref.
Primary	Paclitaxel	Taxanes	Inhibit the proliferation of cancer cells by promoting the polymerization of microtubules in cells	[11]
	Belinostat	Histone deacetylase inhibitor (HDACi)	Epidrug; Treatment of peripheral T-cell lymphoma	[12]
	5-azacitidine	DNA methyltransferase inhibitor (DNMTi)	Epidrug; Treatment of myelodysplastic syndrome	[12]
	Tocilizumab (actemra)	Monoclonal Ab	IL-6 inhibitors; Inhibit the formation of cancer stem cells in TNBC cells	[13]
Metastasis	Doxorubicin	Anthracyclins	Interfering with direct apoptotic effects and DNA local isomerase II action	[14]
	Cyclophosphamide	Alkylating agents	Changes in active or inactive metabolites in the body and anticancer effects of phosphoramide mustard	[15]
	Vorinostat	Histone deacetylase inhibitor (HDACi)	Epidrug; Treatment of cutaneous T-cell lymphoma	[12]
	Zemetostat	Histone methyltransferase inhibitor (HMTi, EZH1i)	Epidrug; Treatment of epithelioid sarcoma, relapsed or refractory follicular lymphoma	[16]
	Atezolizumab +Paclitaxel	Monoclonal Ab	Prolongs progression-free survival in patients with PD-L1-positive breast cancer; Inhibition of metastatic TNBC targeting PD-L1	[17]
	Methotrexate	Antimetabolites	Inhibit nucleotide synthesis; increase AICAR concentration	[18]
	Tivozanib	Tyrosine kinase inhibitor (TKI)	Inhibitors of VEGF/VEGFR; Inhibits angiogenesis	[19]
	Trastuzumab	Monoclonal antibody	Targets the HER2/neu receptor on cancer cells	[20]
	Lapatinib	Tyrosine kinase inhibitor (TKI)	Signal transduction inhibitors of epidermal growth factor receptor (EGFR) and human epidermal receptor type 2 (HER2)	[21]
	Palbociclib	Antineoplastic agents	CDK4/6 inhibitor; prevent cells from moving from the G1 to the S cell cycle phase during division	[22]
	Pertuzumab	Monoclonal antibody	Targets the HER2/neu receptor; induce antibody-dependent cell-mediated cytotoxicity (ADCC)	[23]
	Epirubicin	Anthracyclines	An epimer of doxorubicin; blocks the synthesis of nucleic acids and proteins by inserting planar rings between nucleotide base pairs	[24]

## Data Availability

Not applicable.

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
