# Peer review of "Breast Cancer Metastasis: Mechanisms and Therapeutic Implications"

_ijms, 2022, doi:10.3390/ijms23126806_

Round 1

Reviewer 1 Report

The authors Misung Park, Dohee Kim, Sunghyub Ko, Ayoung Kim, Kyumin Mo and Hyunho Yoon have submitted a review paper entitled "Breast Cancer Metastasis: Mechanisms and Therapeutic Implications" of a great quality and very well written. The review covers up a lot of information on breast cancer and breast cancer metastasis, from the molecular mechanisms to therapies. 

I only have some minor revision:

Line 24: you wrote "aggression" but did you mean "aggressiveness"?

Line 63: there's a word missing before "poor". Did you mean "associated with poor clinical outcomes"?

Line 110: you wrote "mesenchymal-to-epithelial metastasis" but you mean "mesenchymal-to-epithelial transition". 

Line 117: you wrote "lymphocytes" but you mean "lymphatic vessels".

Line 117: you wrote "cause MET", but it is not like that. Cancer cells do not cause MET, they undergo MET. Please correct.

Line 119: you wrote "and migrate to distant sites" but it is not correct. These cells have already migrated distant sites, they have already migrated through blood and lymphatic vessels, and once in the metastasis site, they undergo MET, not to migrate but to settle. Please correct.

Line 168: You wrote "drivers" but you mean "markers". Please correct.

Therefore, considering the great work of the authors, I would recommend this manuscript for publication after correction of these minor revision.

Reviewer 2 Report

Summary:

In the current review “Breast Cancer Metastasis: Mechanisms and Therapeutic Implications” authors intend to shed a light on the various mechanisms leading to metastasis and current therapies to improve the early diagnosis and prognosis in patients with metastatic breast cancer. Manuscript is written in a decent way; however, I have the following comments:

1.    By now, there are more receptor targets identified and their inhibitors being used in the clinical trials, why authors have shown only three signaling receptors and their inhibitors in Figure 1. Is there any specific reason? If not, please update the figure and text accordingly.

2.      Throughout the text, gene names must be italicized.

3.      Few places English may be improved i.e., line 266; line 392-394 (under SALL2 text section).

4.      Line 238, abbreviation is written incorrectly, it should be MDSCs.

5.      The ending conclusive line/s under some sections need to be refined.
